# SPARLING: LEARNING LATENT REPRESENTATIONS WITH EXTREMELY SPARSE ACTIVATIONS

## ABSTRACT

Real-world processes often contain intermediate state that can be modeled as an extremely sparse tensor. We introduce SPARLING, a technique that allows you to learn models with intermediate layers that match this state from only end-to-end labeled examples (i.e., no supervision on the intermediate state). SPARLING uses a new kind of informational bottleneck that enforces levels of activation sparsity unachievable using other techniques. We find that extreme sparsity is necessary to achieve good intermediate state modeling. On our synthetic DIGITCIRCLE domain as well as the LaTeX-OCR and AUDIOMNISTSEQUENCE domains, we are able to precisely localize the intermediate states up to feature permutation with $> 90\%$ accuracy, even though we only train end-to-end.

## 1 INTRODUCTION

A hallmark of deep learning is its ability to learn useful intermediate representations of data from end-to-end supervision via backpropagation. However, these representations are often opaque, with components not referring to any semantically meaningful concepts. Many approaches have been proposed to address this problem. For instance, concept bottlenecks leverage labels for the intermediate concepts (Koh et al. (2020)), and information bottlenecks require that that the mutual information between the representation and the input be bounded (Bourlard & Kamp (1988)). Here, we consider the constraint of extreme sparsity, which, when applicable, leads to a particularly effective approach to discovering the true underlying structure purely by training on end-to-end data.

We introduce SPARLING, a novel technique for learning *extremely sparse representations*, where $\geq 99\%$ of the activations are sparse for a given input. We are motivated by settings where components of the intermediate representation correspond to spatial concepts—which we call *motifs*—that occur in only a small number of locations. For instance, in a character recognition task, each motif may encode whether the center of a given character occurs at a given position. Since even in the worst case, an image of pure text, the image has orders of magnitude fewer characters than pixels, we expect the intermediate representation to be extremely sparse. This pattern is representative of many other prediction tasks—e.g., one could predict economic signals from satellite data by identifying a small number of building types, or transcribe sentences by reading words from noisy audio.

SPARLING directly enforces sparsity by setting activations below some threshold equal to zero; this threshold is iteratively updated to achieve a target sparsity level (e.g., 99%). A key challenge is that the optimization problem is very unstable for high sparsity values. To address this issue, our optimization algorithm anneals the target sparsity over time. A byproduct of this approach is that we achieve a tradeoff between sparsity values and accuracies during the course of training, enabling the user to post-hoc choose a desired sparsity level.

**Example.** Figure 1 shows a sample task we call DIGITCIRCLE. The input consists of noisy images that contain digits placed in a circle, and the output is a list of the digits in counterclockwise order starting from the smallest one.

In addition to solving the end-to-end task, our goal is to train a network with an intermediate layer that precisely identifies the positions of the individual digits in the image. Such an intermediate layer might be useful as a starting point for other tasks that require identifying digits. The challenge is to discover it in the absence of any training data labeling the individual digit locations.

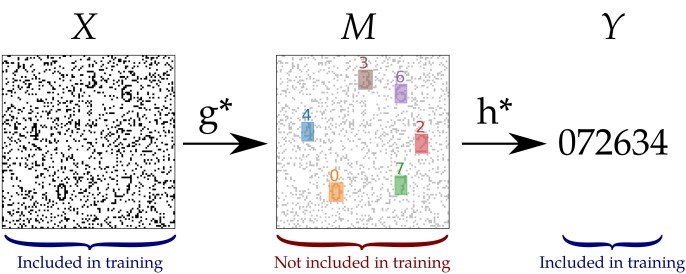

Figure 1: Example of the DIGITCIRCLE domain. The input $x$ is mapped by the ground truth $g^*$ function to a map $m$ of the positions of every digit, which is itself mapped by the ground truth $h^*$ function to the output $y$, the sequence `072634`. Only $x$ and $y$ are available during training.

SPARLING is able to do this by exploiting the expectation of sparsity in this intermediate representation: we know the representation will be extremely sparse because the image contains many pixels but only a small number of digits. Specifically, our training technique is able to achieve nearly the maximum sparsity possible (99.9950%; the maximum sparsity possible for this domain is 99.9955%). Enforcing this level of sparsity forces the representation to identify the correct digit positions 98.84% of the time on average (up to permutation) while achieving end-to-end accuracy of 99.26%. Alternate sparsity enforcement techniques employing $L_1$ and KL-divergence loss either do not produce extreme sparsity or have accuracy below 50%, and as a result, do not lead to intermediate layers that identify individual digits.

**Contributions.** We contribute a new understanding of how enforcing extreme sparsity on an intermediate representation can be used to discover underlying structure. Additionally, we contribute SPARLING, an algorithm for learning intermediate representations with extremely sparse activations, along with an empirical evaluation of the effectiveness of our approach. In particular, we demonstrate that the high motif accuracy from end-to-end training illustrated in DIGITCIRCLE can be achieved on two other, more realistic, domains: LATEX-OCR, in which we predict a LaTeX sequence from a noisy image of an algebraic expression, and AUDIOMNISTSEQUENCE, in which we predict a number from noisy audio of digits being spoken. We will follow DIGITCIRCLE throughout the paper but present motif and end-to-end accuracy results on all 3 domains.

## 2 RELATED WORK

**Concept bottleneck models.** There has been work on learning models with intermediate features that correspond to known variables. Some techniques, such as Concept Bottleneck Models (Koh et al. (2020)) and Concept Embedding Models (Zarlenga et al. (2022)), involve additional supervision with existing feature labels. Other techniques, such as Cross-Model Scene Networks (Aytar et al. (2017)), use multiple datasets with the same intermediate representation. SPARLING does not require the presence of additional datasets or annotations.

**Neural Input Attribution.** SPARLING is useful for identifying the relevant parts of an input. One existing technique that accomplishes this goal is saliency mapping (Simonyan et al. (2013); Selvaraju et al. (2016)), which uses gradient techniques to find which parts of the input affect the output most. Another technique, analyzing the attention weights of an attention layer (Mnih et al. (2014)), only works with a single layer of attention and does not necessarily produce valid or complete explanations (Serrano & Smith (2019)). The main benefit a sparse annotation provides over these techniques is unconditional independence: when using sparsity, you have the ability to make the claim "region $x[r]$ of the input is not relevant to the output prediction, regardless of what happens in the rest of the input $x[\bar{r}]$". This is a direct result of sparsity and locality and is unavailable when using saliency or attention techniques which inherently condition on the values you provide for $x[\bar{r}]$.

**Latent ground truth.** While deep neural networks typically have inscrutable latent variables that are not intended to correspond to any understood feature, in other settings, such as graphical models, latent variables can often represent known quantites. For example, Hidden Markov Models are

commonly used in genomics (Yoon (2009)), where hidden states represent various hidden features of an observed DNA or RNA sequence. Our work attempts to accomplish the same goal of having an interpretable latent variable, but without having to pre-specify what it means.

**Disentangled representations.** *Disentangled representations* are ones where different components of the representation encode independent attributes of the underlying data (Desjardins et al. (2012); Higgins et al. (2016)). Some work suggests there are no universal solutions to this problem, and all attempts require some prior about the kinds of representations being disentangled (Locatello et al. (2019)). We focus here on a prior regarding sparsity and locality.

**Informational bottleneck.** Other work also constrains the information content of the intermediate representation in a neural network. Intuitively, by limiting the mutual information between the input and the the intermediate representation, the model must learn to compress the input in a way that retains performance at the downstream prediction task. Strategies include constraining the dimension of the representation—e.g., PCA and autoencoders with low-dimensional representations (Bourlard & Kamp (1988)), or adding noise—e.g., variational autoencoders (Kingma & Welling (2014)). However, these approaches do not always learn interpretable representations of an intermediate state, as they can encourage entangling features to communicate them through a smaller number of channels.

**Sparse activations.** Note that this notion of sparsity differs from *sparse parameters* (Tibshirani (1996); Scardapane et al. (2017); Ma et al. (2019); Lemhadri et al. (2021)); instead this line of work attempts to constrain the information content of an intermediate representation by encouraging *sparse activations*—i.e., each component of the representation is zero for most inputs. Strategies for achieving sparse activations include imposing an $L_1$ penalty on the representation or a penalty on the KL divergence between the representation's distribution and a low-probability Bernoulli distribution (Jiang et al. (2015)). However, these techniques typically only achieve 50%-90% sparsity, whereas SPARLING can achieve $> 99.9\%$. We directly compare with these in Section 5.1. Additionally, Bizopoulos & Koutsouris (2020) uses a quantile-based activation limit equivalent to both of our ablations (see Section 5.2) combined, but in the simpler context of linear $h$ and $g$ models.

## 3 PRELIMINARIES

We are interested in settings where the activations are latent variables corresponding to semantically meaningful concepts in the prediction problem. To this end, we consider the case where the *ground truth* is represented as a function $f^* : X \to Y$ composed $f^* = h^* \circ g^*$ of two functions $g^* : X \to M$ and $h^* : M \to Y$. Our goal is to learn models $\hat{g}$ and $\hat{h}$ that model $g^*$ and $h^*$ well using only end-to-end data, i.e., enforcing only that their composition $\hat{f} = \hat{h} \circ \hat{g}$ models $f^*$ well.

We assume that elements of $X$ are tensors $\mathbb{R}^{d_1 \times \ldots \times d_k \times C}$, and $Y$ is an arbitrary label space. We typically think of the $C$ dimension as a channel dimension and $d_1 \ldots d_k$ as spatial dimensions (e.g., 2D images). We call the latent space $M$ the *motif* space. We assume it shares spatial dimensions with $X$, but may have a different number of channels. Importantly, we do not assume that $M$ is known—we may have little or no labeled data on which components of $M$ are active.

### 3.1 MOTIF IDENTIFIABILITY

For our approach to work, we require that the output of $g^*$ is *sparse*, that $g^*$ is *local*, and that $g^*$ is *necessary*. We define a motif model $g$ as sparse if its *density* $\delta_g$ is low, ideally close to the minimum necessary for a task. We define $\delta_g$ to be the mean fraction of output activations $g(x)$ that are nonzero. Locality is the standard property where a component only depends on a small number of inputs — e.g., convolutional layers are local. Necessity is the property that $g^*$ only encodes aspects of the input necessary for computing $f^*$—intuitively, if $g^*$ outputs information that is never used by $h^*$ to compute its output, we can not hope to recover that information from end-to-end data on $f^*$.

While these constraints may appear strict, they fit problems where $g^*$ identifies small local patterns in the input—e.g. motifs such as the individual digits in DIGITCIRCLE—and $h^*$ computes something using all the information in those motifs. In these settings, we argue that Motif Identifiability is possible. Specifically, we claim that if $g^*$ and $\hat{g}$ both satisfy sparsity, locality, and necessity, and $\hat{f} \approx f^*$, we can then conclude that $\hat{g} \approx_m g^*$. This claim implies that, for certain kinds of functions,

it is possible to recover the underlying motifs with just end-to-end data. This is an empirical claim, which we validate in our experiments.

## 3.2 MOTIF MODEL EQUIVALENCE AND EVALUATION METRICS

Evaluating Motif Identifiability requires a definition of approximate equivalence between motif models—i.e., what $\hat{g} \approx_m g^*$ means. In particular, the definition of equivalence needs to account for channel permutations and motif alignment. For permutations, the learned motifs model should be deemed equivalent to the ground truth model—which is known when we test—if there exists a channel permutation for which they are equivalent. Motif alignment is the problem that the exact "position" of a motif is arbitrary. To ensure we account for this flexibility when evaluating models, we only check that the predicted point be within the *footprint* of the true motif, which we define as the smallest cuboid[1] covering the points in the motif.

Next, we describe the metrics we use to evaluate different models $\hat{f} = \hat{g} \circ \hat{h}$. First, we use edit (Levenshtein) distance divided by length as our reported metric of end-to-end error:

$$\text{E2EE}_{\mathcal{D}}(\hat{f}) = \mathbb{E}_{x \sim \mathcal{D}} \left[ \frac{\text{EDITDISTANCE}(f^*(x), \hat{f}(x))}{\max(|f^*(x)|, |\hat{f}(x)|)} \right].$$

This error metric can be calculated given only end-to-end supervision in the form of $(x, y)$ pairs. We then define three motif error metrics to evaluate $\hat{g} \approx_m g^*$; these metrics are only used for testing purposes, since they assume knowledge of the true motifs.

First, the *false positive error (FPE)* is the percentage of motifs that are false positive motifs.

$$\text{FPE}_{\mathcal{D}}(\hat{g}) = \frac{\sum_{x \in \mathcal{D}} |\text{FPM}(\hat{g}(x), g^*(x))|}{\sum_{x \in \mathcal{D}} |P(\hat{g}(x))|}.$$

We define $P(\hat{m})$ as the set of all predicted motifs (PMs), and $\text{FPM}(\hat{m}, m^*)$ is the set of PMs that do not overlap the footprints of any true motifs. Second, the *false negative error (FNE)* is the percentage of true sites that are not covered by any motif. Finally, the *confusion error (CE)* is defined as follows: (i) permute $\hat{g}$'s channels to best align them with $g^*$, (ii) compute the percentage of maximal motifs in footprint of a true motif that do not correspond to the true motif's channel:

$$\text{CE}_{\mathcal{D}}(\hat{g}) = \min_{\sigma \in \Sigma_C} \frac{\sum_{x \in \mathcal{D}} |\text{conf}_\sigma(\hat{g}(x), g^*(x))|}{\sum_{x \in \mathcal{D}} |\text{MM}(\hat{g}(x), g^*(x))|},$$

where $\text{MM}(\hat{m}, m^*)$ is the set of PMs that overlap a footprint of a true motif and have greater activation value than all other motifs overlapping the same footprint.[2] $\text{conf}_\sigma(\hat{m}, m^*)$ represents the motifs that do not match ground truth under permutation $\sigma$

$$\text{conf}_\sigma(\hat{m}, m^*) = \{t \in \text{MM}(\hat{m}, m^*) : \neg \text{mat}_\sigma(t, C(t, m^*))\}|,$$

$\text{mat}_\sigma(\hat{t}, t^*)$ is a function that checks whether the two motif index tuples match under channel permutation $\sigma$, and $C((\hat{\mathbf{i}}, \hat{c}), m^*)$ is the footprint that the predicted motif at location $\hat{\mathbf{i}}, \hat{c}$ matches, or $\emptyset$ if it does not match any.

A low FPE/FNE implies that the model is identifying relevant portions of the input, while a low CE implies that the model classifies these components as motifs correctly. Appendix A contains formal definitions of the functions not defined formally here.

## 3.3 CONNECTION TO INFORMATION BOUND

Sparsity induces an information bound by limiting the amount of information in the intermediate representation. Specifically, if we let $\mathcal{X}$ be a random variable for the input, and $\mathcal{M} = g(\mathcal{X})$ be the motif layer, we have that we can bound the mutual information between inputs and motifs as $I(\mathcal{X}, \mathcal{M}) \leq H(\mathcal{M})$, where $H(\cdot)$ is entropy. Thus, to bound mutual information, it is sufficient to

---

[1]For images, the cuboid is a rectangle, as drawn in Figure 1, for audio the cuboid is a temporal interval.

[2]We ignore motifs not maximal in a footprint as these could be filtered by a user of the motif predictions.

---

**Algorithm 1** Train Loop $(\hat{f}, \mathcal{D}, M, B, d_T, \delta_{\text{update}})$

---

$T_0 \leftarrow 1$
**for** $t = 1$ **to** ... **do**
    $\textsc{TrainStep}(\hat{f}, \mathcal{D}_{Bt:B(t+1)})$
    $T_t \leftarrow T_{t-1} - B d_T$
    **if** $bt \bmod M = 0$ **then**
        $A_t \leftarrow \textsc{Validate}(\hat{f})$
        **if** $A_t > T_t$ **then**
            $\hat{f}.\delta \leftarrow \hat{f}.\delta \times \delta_{\text{update}}$
            $T_t \leftarrow A_t$

---

bound $H(\mathcal{M})$. We first can break it into per-channel components: $H(\mathcal{M}) \leq \sum_{\mathbf{i},c} H(\mathcal{M}[\mathbf{i}, c])$, Then, let $\delta_{\mathbf{i},c}$ denote the density of channel $c$ at position $\mathbf{i}$, and $\eta \geq H(\mathcal{M}[\mathbf{i}, c] | \mathcal{M}[\mathbf{i}, c] \neq 0)$ be a bound on the amount of entropy in each nonzero activation (see Appendix B). Then we apply the chain rule to get $H(\mathcal{M}[\mathbf{i}, c]) \leq H(B(\delta_{\mathbf{i},c})) + \eta \delta_{\mathbf{i},c}$ where $B(\cdot)$ is the Bernoulli distribution. Thus, $H(\mathcal{M}) \leq \sum_{\mathbf{i},c} H(B(\delta_{\mathbf{i},c})) + SC\eta\delta$, where $S$ is the size of the image in pixels and $C$ is the number of channels, and $\delta$ is defined as in section 3.1. Finally, using Jensen's inequality (as $H(B(t))$ is concave):

$$I(\mathcal{X}, \mathcal{M}) \leq H(\mathcal{M}) \leq SC(H(B(\delta)) + \eta\delta).$$

This demonstrates that a sparsity bound can be used as an information bound and thus that SPARLING operates as a kind of informational bottleneck.

## 4 METHODS

In this section, we introduce SPARLING, which is composed of two parts: the Spatial Sparsity Layer and the Adaptive Sparsity Algorithm. The Spatial Sparsity Layer is designed to achieve the extreme sparsity levels described in Section 3. This layer is the last step in the computation of $\hat{g}$ and enforces the sparsity of $\hat{g}$; we compose $\hat{g}$ out of convolutional layers to enforce locality. The Adaptive Sparsity Algorithm is designed to ensure the Spatial Sparsity Layer can be effectively trained.

### 4.1 SPATIAL SPARSITY LAYER

We define a spatial sparsity layer to be a layer with a parameter $t$ whose forward pass is computed

$$\text{Sparse}_t(z) = \text{ReLU}(z - t)$$

Importantly, $t$ is treated as a constant in backpropagation and is thus not updated by gradient descent. Instead, we update $t$ using an exponential moving average of the quantiles of batches[3]:

$$t_n = \mu t_{n-1} + (1 - \mu) q(z_n, 1 - \delta),$$

where $t_n$ is the value of $t$ on the $n$th iteration, $z_n$ is the $n$th batch of inputs to this layer, $\mu$ is the momentum (we use $\mu = 0.9$), $\delta$ is the target density, and $q : \mathbb{R}^{B \times d_1 \times ... \times d_k \times C} \times \mathbb{R} \to \mathbb{R}^C$ is the standard `torch.quantile` function. $q$ is applied across all dimensions except the last: it produces a value for each channel that represents the threshold $u$ for which the proportion of elements above $u$ in the tensor at that channel is $\delta$. We describe an alternative in Appendix D. Since $t_n$ is fit to the data distribution, we can treat this as a layer that enforces that $\hat{g}$ has a sparsity of $1 - \delta$.

Finally, we always include an affine batch normalization before this layer to increase training stability. We provide an analysis on the necessity of this addition in Section 5.2.

### 4.2 ADAPTIVE SPARSITY

In practice, we find that applying an extreme sparsity requirement (very low $\delta$) upon initial training of the network leads to bad local minima, with the network being unable to gain any learning signal on the vast majority of inputs. Instead, we use a technique inspired by simulated annealing and

---

[3]For numerical stability, we accumulate batches such that $|z_n|\delta \geq 10C$ before running this update

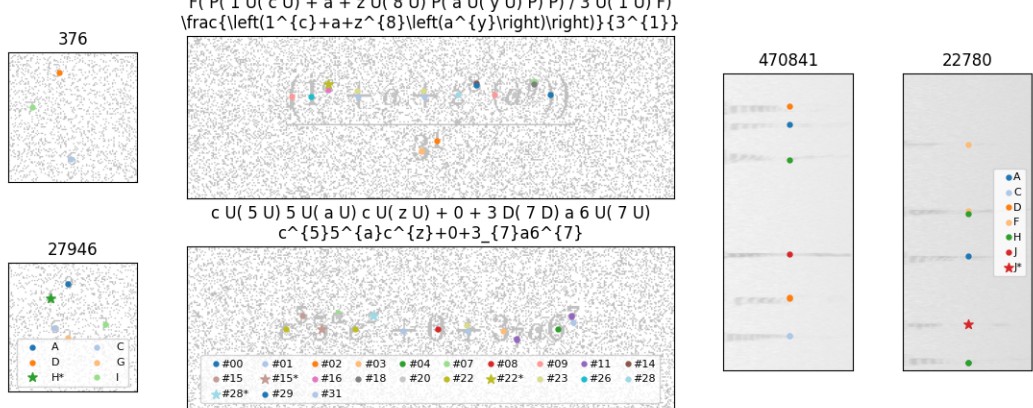

Figure 2: Examples of input/output pairs of our domains, along with the model's motif predictions. The inputs are the images, and outputs are the sequences in the titles. For LaTeX-OCR, we provide the output twice, first as the sequence of commands generated by the network and second as the translation of those commands into LaTeX. We annotate in dots the maximal motifs produced by the $\hat{g}$ of the model with seed=1. We label our activations A through J or with number signs to distinguish them from digits. Stars indicate sites where there are non-maximal motifs present as well. Note that in the LaTeX domain, the symbols + ( ) are not always recognized and that the fraction bar is never recognized: this is because these motifs are not needed to infer the LaTeX output.

learning rate decay, and reduce $\delta$ slowly over time. Annealing hyperparameters is a known technique Sønderby et al. (2016), but we tie this annealing to validation accuracy (we use exact match between $y$ and $\hat{y}$) in order to automatically tune it and avoid introducing an additional hyperparameter.

Specifically, we add a step to our training loop that periodically checks validation accuracy $A_t$ and reduces the density whenever it exceeds a target $T_t$. The process is as described in Algorithm 1, with the target accuracy dropping slowly. When the validation accuracy reaches the target accuracy, we reduce density and increase the accuracy bar to whatever our model achieved. Our experiments use evaluation frequency $M = 2 \times 10^5$, batch size $B = 10$, $d_T = 10^{-7}$, and $\delta_{\text{update}} = 0.75$.

## 5 EXPERIMENTS

### 5.1 EXPERIMENTAL SETUP

We have three domains, as described below. See Figure 2 for examples of each domain.

**DIGITCIRCLE domain.** To evaluate SPARLING we construct the DIGITCIRCLE domain. The input $X$ is a $100 \times 100$ monochrome image with 3-6 unique digits placed in a rough circular pattern, with some noise being applied to the image both before and after the numbers are placed. The output $Y$ is the sequence of digits in counterclockwise order, starting with the smallest number. The latent motifs layer $M$ is the position of each digit: we can conceptualize this space as a $100 \times 100 \times 10$ tensor with 3-6 nonzero entries. Note that the model during training and validation has no access to the concept of a digit as an image, nor to the concept of a digit's position.

**LaTeX-OCR domain.** To provide a more realistic test of our model's capabilities we use the task of synthesizing LaTeX code from images. This task is similar to DIGITCIRCLE in that the motifs are digits, but these digits vary in size and pixel-level rendering, and the motif-to-output relationship is also more complex. This task is inspired by Deng et al. (2016).

**AUDIOMNISTSEQUENCE domain.** In this domain, we synthesize short clips of audio representing sequences of 5-10 digits over a bed of noise. The task is to predict the sequence of characters spoken. Here, we test if motif models can generalize: we train and validate with AUDIOMNIST (Becker et al. (2018)) samples from Speakers 1-51 and test with samples from Speakers 52-60.

Figure 3: Motif Error. Bar height depicts the mean across 9 seeds, while individual dots represent the individual values and the error bar represents a 95% bootstrap CI of the mean. AUDIOMNISTSE-QUENCE has an FPE of exactly 0, so it does not appear on the graph. High FNE on LaTeX-OCR is due to fraction bars, parentheses, and plus signs not being recognized in all cases since it is possible to infer the LaTeX code without access to these.

**Architecture and training.** Our neural architecture is adapted from that of Deng et al. (2016). For DIGITCIRCLE, we make $\hat{g}$ have a $17 \times 17$ overall window, by layering four residual units (He et al. (2016)), each containing two $3 \times 3$ convolutional layers. We then map to a 10-channel bottleneck where our Spatial Sparsity layer is placed. (We choose 10 channels to match the 10 digits.) Our $\hat{h}$ architecture is a max pooling, followed by a similar architecture to Deng. We keep the LSTM row-encoder, but replace the attention decoder with a column-based positional encoding followed by a Transformer (Vaswani et al. (2017)) whose encoder and decoder have 8 heads and 6 layers. Throughout, except in the bottleneck layer, we use a width of 512 for all units. For LaTeX-OCR we use the same architecture but with 32 motifs (to account for the additional characters) and a $65 \times 65$ overall window (to account for the larger characters)[4]. For AUDIOMNISTSEQUENCE we use a modification of this architecture designed for 1-dimensional data, processing the audio via a spectrogram with a sample rate of 8000 and 64 channels, and treating the signal as a 1-dimensional signal from there. We use a 33-wide 1D convolutional filter for the motifs and remove the LSTM from the post-sparse model, making it entirely a transformer. For comparisons to baselines and ablations we keep the model architecture fixed and only modify the Sparse layer.

We generate training datasets randomly, with seeds 1 through 9 for the 9 different training runs of each model, and seeds -1 and -2 being reserved for validation and testing. For efficiency, LaTeX-OCR is limited to $10^7$ training samples, after which it repeats. We use a batch size of 10 samples and a learning rate of $10^{-5}$. Our validation and test sets both contain $10^4$ examples.

**Baselines.** We consider two other approaches to ensuring the creation of sparse motifs, both taking the form of auxiliary regularization losses. In both cases, we vary loss weight to analyze how that affects error and sparsity. First, we consider $L_1$ loss. In our implementation, we use an affine batch normalization layer followed by a ReLU. The output of the ReLU is then used in an auxiliary $L_1$ loss[5]. This approach is discussed in Jiang et al. (2015). We also consider using $KL$-divergence loss as in Jiang et al. (2015). The approach is to apply a sigmoid, then compute a KL-divergence between the Bernoulli implied by the mean activation of the sigmoid and a target sparsity value (we use 99.995% to perform a direct comparison). While this usually is done across the training data Ng (2011), we instead enforce the loss across all positions and channels, but per-batch (the mean sparsity should be similar in each batch). Our other modification, in order to induce true sparsity, is to, after the sigmoid layer (where the loss is computed), subtract 0.5 and apply a ReLU layer.

**Ablations.** We consider two ablations: First, is the batch normalization we place before our sparse layer necessary? Second, is the adaptive sparsity algorithm we use necessary? These ablations are only evaluated on DIGITCIRCLE as it is the domain where simpler techniques would work best.

## 5.2   RESULTS

**Motif error.** We show our metrics of motif error, FNE, FPE, and CE in Figure 3 for each of our models on each domain. Motif errors for our model average below 10% for all our domains, except in the case of FNE on LaTeX-OCR. The generally low motif errors, despite only training and validating end-to-end, demonstrate that our algorithm achieves Motif Identifiability on all three

---

[4]In practice we find that using $33 \times 33$ does not substantially change the results

[5]This approach parameterizes the same model class as SPARLING; both act as a ReLU in a forward pass

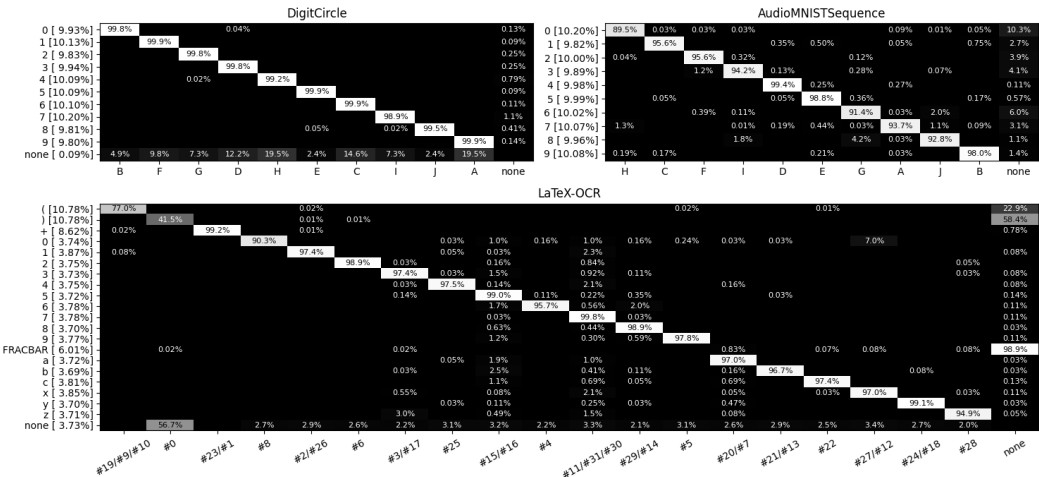

Figure 4: Confusion Matrix of 10k unseen samples computed for seed=1 across all domains. False positive and false negative motifs are placed into the `none` rows and columns, respectively. Each row is labeled by the percentage of motifs falling into the row, and each row's cells are then normalized to add to 1. We then permute to align along the diagonal. For LATEX-OCR, we use more channels than there are symbol types so we merge channels together for display and analysis.

domains. This property even holds when generalizing to unseen samples in the AUDIOMNISTSE-QUENCE experiment, providing evidence that SPARLING is genuinely learning the motif features rather than memorizing. The one case where our model has high error, FNE on LATEX-OCR, demonstrates the importance of the Necessity Assumption: recognizing LATEX text in the space we generated does not require identification of fraction bars or all of `()+`. For more details, see Figures 2 and 4.

**Examples.** Figure 2 shows a few examples for one of our models' intermediate layers. As can be seen, all digits are appropriately identified by our intermediate layer, with very few dots (in these examples, none) falling away from a digit. Note that the activations are consistent from sample to sample—for example, in DIGITCIRCLE, motif C is used for digit 6 in both images.

**Confusion matrices.** Figure 4, depicts appropriately permuted confusion matrices for each domain. Our model generally assigns each true motif to a channel or set of channels in the sparse layer. The main exception is that in LATEX-OCR, the fraction bar is never recognized, and `()` are only sometimes recognized. In other seeds, `+` exhibits similar behavior to `()`.

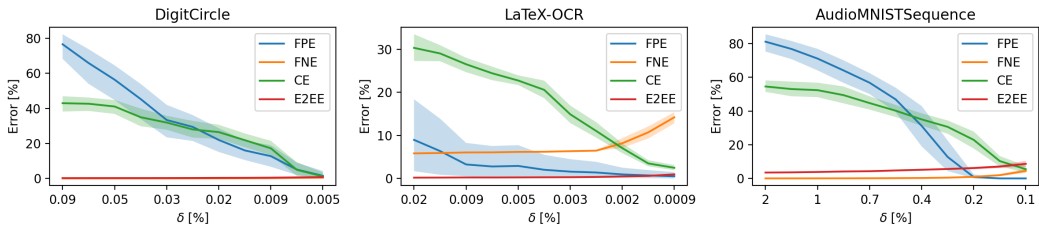

Figure 5: Motif and end-to-end error metrics versus $\delta$. Note that the $x$ axis is reversed, this is to indicate training progression, which starts with high density and narrows it over time.

**Necessity of Extreme Sparsity** Figure 5 shows our error metrics plotted against the sparsity, with the $x$-axis reversed to show progression in training time as we anneal $\delta$. As expected, as $\delta$ decreases, FPE decreases and FNE increases. More interestingly, we note a tradeoff between E2EE and CE: as $\delta$ decreases, E2EE increases and CE decreases substantially. This demonstrates a tradeoff between a more accurate overall model, which benefits from greater information present and a more accurate

| | $L_1$ | | | | | SPARLING |
| | $\lambda = 0.1$ | $\lambda = 1$ | $\lambda = 2$ | $\lambda = 5$ | $\lambda = 10$ | MT |
|---|---|---|---|---|---|---|
| FPE [%] | 99.99 | 99.90 | 91.25 | 95.99 | 97.63 | 1.48 [0.07-4.23] |
| FNE [%] | 0.00 | 0.00 | 58.09 | 73.12 | 84.51 | 0.42 [0.25-0.67] |
| CE [%] | 50.34 | 47.84 | 45.65 | 50.85 | 33.82 | 1.16 [0.03-3.39] |
| E2EE [%] | 0.68 | 2.85 | 70.31 | 75.00 | 73.20 | 0.74 [0.47-1.15] |
| Density [%] | 37 | 4.7 | 0.023 | 0.032 | 0.028 | 0.005 |

Table 1: Results of $L_1$ experiment on DIGITCIRCLE. As $L_1$ increases, the density decreases, but end-to-end error becomes $> 50\%$, and CE/FPE never improve to the level of SPARLING. SPARLING is able to keep error low while achieving lower density than $L_1$ with any $\lambda$ value we tried.

motif model, which benefits from a tighter entropy bound. Furthermore, CE is often substantially higher for even a 2-3$\times$ increase in $\delta$, demonstrating the need for extreme sparsity.

**Baselines.** Table 1 shows the results of using $L_1$ as a method for encouraging sparsity. There are two weight regimes, where when $\lambda \leq 1$, we end up with high density (relative to the theoretical minimum) but low error, and when $\lambda \geq 2$, we end up with high-error model. Even in the latter case, the $L_1$ loss does not consistently push density down to the level of SPARLING, suggesting it might be insufficiently strong as a learning signal. In our experiments, the $KL$-divergence was unable to achieve a density below 0.1%, even when we used a loss weight as high as $\lambda = 10^5$ and $3 \times 10^6$ steps (much more than was necessary for convergence of the $L_1$ model). Thus, we conclude that it is unsuitable for encouraging the kind of sparsity we are interested in.

**Ablation** We compare our approach to ablations to evaluate our design decisions. First, including a batch normalization before the sparsity layer is crucial. Without a batch normalization layer, over 9 runs, the best model gets an E2EE of 71%, in essence, it is not able to learn the task at all. Additionally, annealing (Algorithm 1) is clearly necessary: when started with the annealing algorithm's final and penultimate $\delta$ values, the model converged to E2EE values of 68% and 71% respectively.

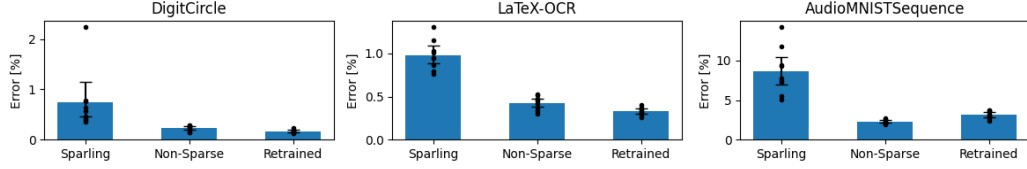

Figure 6: *Retrained* tends to perform as well as or slightly worse than *Non-Sparse*, making up most of the gap from SPARLING

**End-to-End error** As seen in Figure 6, SPARLING tends to produce higher end-to-end errors than a baseline Non-Sparse model. This is to be expected as we are imposing a constraint on the information flow that requires, the model to "commit" to a choice on whether or not a given site is a true motif. To control for this effect, we present the *Retrained* setting, in which we remove the bottleneck, freeze the motif model $\hat{g}$, and finetune $\hat{h}$ on the training set until convergence. The Retrained setting tends to perform similarly to or only slightly worse than the Non-Sparse setting. We thus demonstrate that we are not losing an unacceptable amount of performance end-to-end even as we are able to substantially improve the interpretability of the model.

## 6 CONCLUSION

SPARLING is a novel spatial sparsity layer and adaptive sparsity training technique that has the ability to learn a highly sparse latent motifs layer for dimensional data using only an end-to-end training signal. Similar levels of activation sparsity are unachievable by existing strategies. We demonstrate that SPARLING achieves interpretable and accurate motifs with zero direct supervision on the motifs. Finally, we demonstrate that SPARLING is not specific to any particular domain: it works well across three different domains.

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

## A  EVALUATION METRIC DETAILS

We now define our FPM and MM motif sets, along with the $C$ function.

**Predicted motifs.** For a given predicted motif tensor $\hat{m}$, we define $P(\hat{m}) = \{(\hat{\mathbf{i}}, \hat{c}) : \hat{m}[\hat{\mathbf{i}}, \hat{c}] > 0\}$ to be the set of motifs predicted in $\hat{m}$, where $\mathbf{i}$ is over all sequences of spatial indices (e.g., for images $\mathbf{i} : \mathbb{N}^2$) and $c$ is over the channel indices. Typically, we are interested in the set of motifs $P(\hat{g}(x))$ for our estimated motif model $\hat{g}$.

**Footprint.** We can formally define the footprint of a motif as follows: Let a motif with $\mathbf{i}$ in channel $c$ have footprint $\mathbf{i} + F_c$. Note that $F_c$ depends on the channel of the true motif—e.g., in the DIGITCIRCLE domain, some digits are slightly larger than others.

**Footprint identification.** First, we define a way to determine which footprint a motif belongs to. Define the footprint motif function $S(\hat{\mathbf{i}}, m^*)$ to be the set of true motifs whose footprints contain $\hat{\mathbf{i}}$—i.e.,

$$S(\hat{\mathbf{i}}, m^*) = \{(\mathbf{i}, c) : m^*[\mathbf{i}, c] \wedge \hat{\mathbf{i}} - \mathbf{i} \in F_c\}.$$

As a simplification, since motif footprints typically do not heavily overlap, we define $C(\hat{\mathbf{i}}, m^*)$ to be our classification function that gives a relevant true motif center for the input $\hat{\mathbf{i}}$.

$$C(\hat{\mathbf{i}}, m^*) = u(S(\hat{\mathbf{i}}, m^*)),$$

where $u$ is a choice function that picks an arbitrary element of its input if there are multiple and returns the empty set if there are no entries.

**False Positive Motifs.** We now have the ability to define our first class of motifs: *false positive motifs*. These are predicted motifs that do not correspond to any real motifs:

$$\text{FPM}(\hat{m}, m) = \{(\hat{\mathbf{i}}, \hat{c}) \in P(\hat{m}) : C(\hat{\mathbf{i}}, g(x)) = \emptyset\}.$$

We denote the remaining motifs by

$$P_1(\hat{m}, m^*) = P(\hat{m}) \setminus \text{FPM}(\hat{m}, m^*).$$

**Maximal Motifs** First, we need to define the set of all predicted motifs that cover the same footprint as a given predicted motif. We do so via the $A_{\hat{m}, m^*}$ function, which takes a given predicted motif (assumed to overlap some footprint) and returns all others covering the same footprint:

$$A_{\hat{m}, m^*}(\hat{\mathbf{i}}, c) = \{(\hat{\mathbf{i}}', \hat{c}') \in P(\hat{m}) : C(\hat{\mathbf{i}}', m^*) = C(\hat{\mathbf{i}}, m^*)\}$$

Now we can define *maximal motifs* are predicted motifs that are maximal in the footprint they cover:

$$\text{MM}(\hat{m}, m^*) = \{t \in P_1(\hat{m}, m^*) : \hat{m}[t] = \max_{t' \in A_{\hat{m}, m^*}(t)} \hat{m}[t']\}$$

We can also define *non-maximal motifs* are predicted motifs that are non-maximal in the footprint they cover:

$$\text{NMM}(\hat{m}, m^*) = \{t \in P_1(\hat{m}, m^*) : \hat{m}[t] \neq \max_{t' \in A_{\hat{m}, m^*}(t)} \hat{m}[t']\}$$

However, we ignore non-maximal motifs entirely for the purposes of our analysis, under the reasoning that these are trivially removable in practice.

## B  ENTROPY UPPER BOUND

To compute our entropy upper bound, we must first compute $\eta$, as defined in Section 3.3. To compute this, we bin the nonzero activations into $2^k$ bins by quantile. We set $\eta$ to be the smallest value of $k$ that does not substantially affect the accuracy of the model (we consider 0.1% to be a reasonable threshold for this purpose). Figure 7 shows the result of this experiment, averaged across 9 seeds. The general downward trend in error caused by binning as density decreases demonstrates that reducing the number of motifs reduces the importance of the precise magnitudes. For the purposes of entropy bounding, we can use $\eta = \log(16) = 4b$.

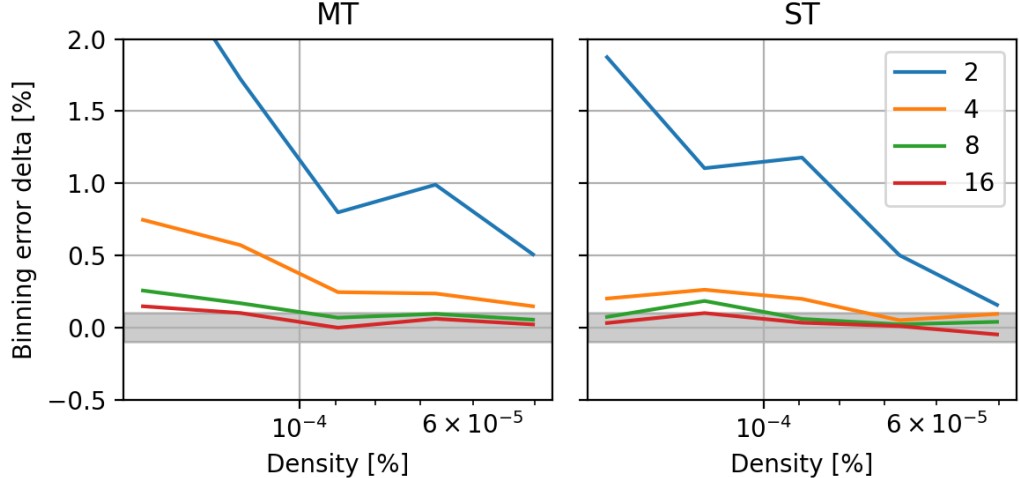

Figure 7: Increase in error when binning. Each series represents a different bin count, as annotated in the legend. Density is log-scaled and reversed to indicate training progress. MT is the model tracked in the rest of the paper, ST is the model as defined in Appendix D

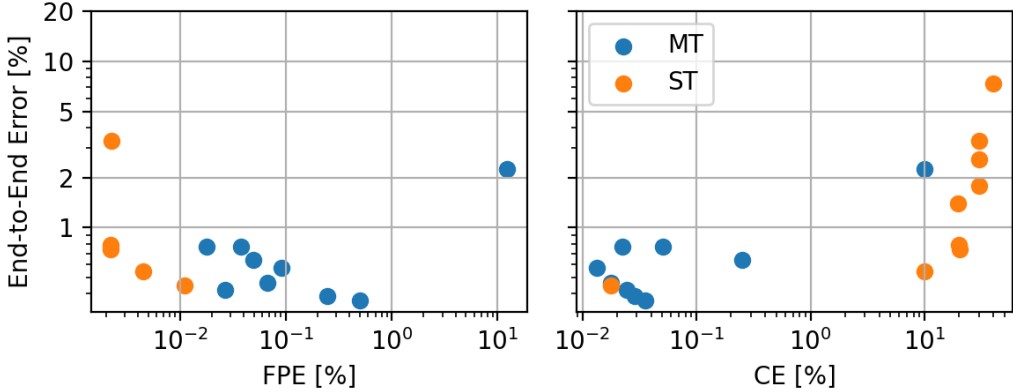

Figure 8: Model error versus FPE and CE, at $1.1\times$ the minimum sparsity. All are log-scaled to highlight the low-error region. Each dot represents a single model training seed.

## C    PREDICTING MOTIF ERROR.

Figure 8 shows the relationship between the motif errors and the overall end-to-end error for DIGIT-CIRCLE. There is no relationship for FPE, but there is a positive relationship for CE, implying that a strategy where one trains several models and then chooses the one with the best validation error is a good way to reduce CE and thereby improve motif quality. This provides further evidence for Motif Identifiability (though the primary evidence for this remains that this model is able to achieve low FPE and CE in general, as training itself focuses on reducing end-to-end error via the loss function). While this may seem to contradict the result in Appendix 5.2, it in fact does not. Within a single model, tightening the density has inverse effects on end-to-end error and CE, but separately, some models are in general more or less accurate.

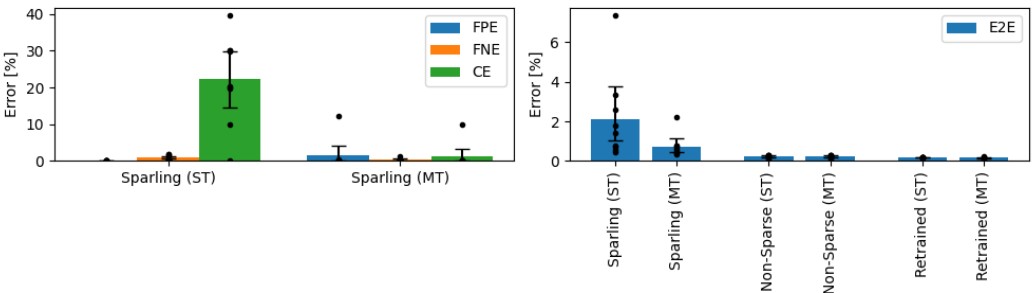

Figure 9: SPARLING using MT (as in the main figures) vs ST

## D  SINGLE THRESHOLD

In this section, we consider a variation to the quantile function. We call this the *single threshold (ST)* sparsity approach, as opposed to the *multiple thresholds (MT)* technique described in Section 4.1, where we take the quantile across the entire input (batch axis, dimensional axes, channel axis). In this case, the channels can have differing resulting densities that average together to the target $\delta$. More precisely, we use the quantile function $q_{\text{ST}} : \mathbb{R}^{B \times d_1 \times \ldots \times d_k \times C} \times \mathbb{R} \to \mathbb{R}$, implemented such that

$$p \approx \frac{1}{BSC} \sum_{b,\mathbf{i},c} \mathbf{1}(z[b,\mathbf{i},c] \leq q_{\text{ST}}(z,p)).$$

As seen in Figure 9, ST performs substantially worse in terms of CE and E2EE, while performing better with respect to FPE. Without the constraint that the motifs have equivalent density across each channel, some motifs are being used to represent multiple digits, which substantially increases confusion error, but also reduces false positives. In general, the MT model is superior as it has reasonable FPE and substantially lower CE/E2EE.

## E  COMPARISON TO DIRECTLY LEARNING THE MOTIFS

|  | SPARLING [mean] | DIRECT [mean] | Ratio [of means] |
|---|---|---|---|
| DigitCircle | 1.24 | 0.01 | 0.01 |
| LaTeX-OCR | 6.55 | 0.12 | 0.02 |
| LaTeX-OCR [without +()] | 2.96 | 0.10 | 0.03 |
| AudioMNISTSequence/train | 5.41 | 0.61 | 0.11 |
| AudioMNISTSequence/test | 8.01 | 4.28 | 0.53 |

Table 2: Error [%] and ratios between errors. All are computed as a mean across 9 seeds

The purpose of SPARLING is to be able to learn intermediate state without having to have access to any training data on the intermediate state. In this section, we analyze how well it does at this goal, by comparing it to DIRECT, a setting where we train and evaluate on the intermediate state directly. Specifically, we construct datasets for each task of single motifs and train and test models on these datasets, then also test SPARLING on these datasets.

In the case of DIGITCIRCLE and LATEX-OCR, DIRECT is a trivial task as there is no distributional shift in the motif samples used to train and evaluate the model – effectively, DIRECT is tested on the training set. Thus, DIRECT gets ∼0% error.

However, on the AUDIOMNISTSEQUENCE task, the DIRECT has non-negligible error, with 0.61% error on the training sample distribution but a much higher 4.28% error on the testing sample distribution. Meanwhile, SPARLING increases substantially less, from 5.41% to 8.01%. This is because the error in SPARLING comes from two sources: the underlying uncertainty in prediction it shares with the DIRECT technique, and epistemic uncertainty related to the problem of identifying motifs

from end-to-end data. This latter error evidently does not scale linearly with the difficulty of the underlying task.

