# OpenReview forum: "Sparling: Learning Latent Representations With Extremely Sparse Activations"
_ICLR.cc/2024/Conference — Submitted to ICLR 2024_

### Official Review · Reviewer_QB7p · 2023-10-27

**Soundness:** 2 fair
**Presentation:** 3 good
**Contribution:** 2 fair
**Rating:** 3
**Confidence:** 3

**Summary:**

The paper looks into the problem of learning sparse intermediate activations that preserve spatial information, which can help interpret the intermediate outcomes better and be useful for certain tasks. The paper argues that existing methods based on regularization, such as L1 norm on activations, have drawbacks of either low sparsity rate or low accuracy. To overcome this, the paper introduces optimizations such as an extra spatial sparsity layer normalization layer before the activation and iteratively induces a higher sparsity rate during training. The paper evaluates the proposed method on three tasks DIGITCIRCLE, LATEX-OCR, AUDIOMNISTSEQUENCE and shows that it can achieve higher sparsity and higher accuracy than several baselines.

**Strengths:**

- The proposed spatial sparsify layer together with the adaptive sparsifying method seems to induce a very high sparsity ratio in activations in the tested tasks.

- The spatial sparsify layer, to the reviewer's best knowledge, seems to be novel.

**Weaknesses:**

1. The work is a bit under-motivated. While the interpretability of deep neural networks is crucial, the paper looks into the problem from the angle of changing the model architecture and training method to obtain more interpretable representations. As such, it is less clear how the proposed method can impact state-of-the-art deep neural networks that have been used in practice. It would be more helpful if the authors could add some real applications where sparse activations are useful.

2. Related to the motivation issue, the datasets the paper uses for evaluation seem to be a bit artificial and at a tiny scale. It would be helpful if the authors could elaborate a bit more on how the tasks/domains can interact with real-world applications. Also, given the small scale of the datasets, it raises questions on how well the proposed methods can generalize to larger and more complex data.

3. The paper introduces many additional hyperparameters, such as M, d_T, /delta_update. However, the paper does not explain how these hyperparameters are selected, such as the search space and the sensitivity of each hyperparameter.

4. The paper claims that the adaptive sparsity training technique is novel, but it seems to be similar to the iterative pruning method proposed in the lottery ticket hypothesis paper, except that the paper applies it to the activation and via the parameter in the spacial sparsify layer. The paper should better clarify the differences.

**Questions:**

Can the authors add some real applications or scenarios where sparse activations are useful?

---

> ### Author Response · Authors · 2023-11-22
>
> > The work is a bit under-motivated. While the interpretability of deep neural networks is crucial, the paper looks into the problem from the angle of changing the model architecture and training method to obtain more interpretable representations. As such, it is less clear how the proposed method can impact state-of-the-art deep neural networks that have been used in practice. It would be more helpful if the authors could add some real applications where sparse activations are useful.
>
> While this technique does require changing the architecture, one important thing to note is that it does not impose any stringent requirements on the architecture of h. In fact, in this work, the architecture of h is a fairly sophisticated one, involving both a LSTM and Transformer. The primary restriction is on the properties of g, which is more of a statement about the domain than about the technique. In future work, we will analyze more realistic applications, but we believe that many tasks have properties similar to the OCR and audio processing tasks we used as validation domains to demonstrate the efficacy of our technique.
>
> > Related to the motivation issue, the datasets the paper uses for evaluation seem to be a bit artificial and at a tiny scale. It would be helpful if the authors could elaborate a bit more on how the tasks/domains can interact with real-world applications. Also, given the small scale of the datasets, it raises questions on how well the proposed methods can generalize to larger and more complex data.
> > Can the authors add some real applications or scenarios where sparse activations are useful?
>
> See https://openreview.net/forum?id=zgHamUBuuO&noteId=tJhJMRIRWh for our response.
>
> > The paper introduces many additional hyperparameters, such as M, d_T, /delta_update. However, the paper does not explain how these hyperparameters are selected, such as the search space and the sensitivity of each hyperparameter.
>
> We did not tune these parameters as the first choices we came up with seemed to work relatively well. If needed, we can perform perturbation analysis of these parameters for a camera ready.
>
> > The paper claims that the adaptive sparsity training technique is novel, but it seems to be similar to the iterative pruning method proposed in the lottery ticket hypothesis paper, except that the paper applies it to the activation and via the parameter in the spacial sparsify layer. The paper should better clarify the differences.
>
> We do not believe there are anything more than superficial similarities between our use of adaptive sparsity and the iterative pruning technique in the Lottery Ticket Hypothesis paper.
> While both involve the increasing of a quantity related to sparsity over the process of training, the goals, practical effect, and implementation, of these operations are drastically different.
>
> In the lottery ticket paper, the enforcement of sparsity is done in order to have the neural network continue to parameterize the same function, while having a more efficient computational graph. We, on the other hand, are attempting to change the function computed by our g and h networks to have a different intermediate latent representation.
>
> The effects are also drastically different. When a neural network has sparse parameters, this is a change to the network, and has the same structure across different inputs and future training steps, Sparse activation, on the other hand, involves modifying the distribution of outputs of a certain layer. Crucially, the pattern of sparsity is not fixed across different inputs, or even across the same input at different points during training. A location that might be activated at one timestep in training might be removed later, even if sparsity does not change. The same can even happen in reverse, with an activation set to 0 in an earlier training iteration set to a value above 0 in a later one.
>
> These differences in effects lead to differences in the technique. We do not assume that we can predict ahead-of-time how the network will react to changes in sparsity, so we use validation loss as a guide to the adaptive sparsity schedule. Additionally, there is no pruning step per se, we merely update the delta value and allow the exponential moving average to compensate over time.
>
> We have added a citation to the lottery ticket hypothesis, in Related Work paragraph “Sparse activations”, as well as some of this explanation.

---

### Official Review · Reviewer_qwvn · 2023-11-01

**Soundness:** 1 poor
**Presentation:** 1 poor
**Contribution:** 2 fair
**Rating:** 1
**Confidence:** 3

**Summary:**

The paper introduces SPARLING, a technique that allows to learn models capable of sparsely identifying structure of data without direct supervision. To do so, the paper formulates the problem of learning sparse tensors with an information bottleneck objective. SPARLING is validated on both synthetic data and visual/audio domains.

**Strengths:**

1. The proposed method is designed to be capable of inducing sparse representations without any direct supervision
2. The evaluation of the proposed algorithm has been conducted on multiple benchmarks from different modalities (providing good clues of the general applicability of the proposed idea).

**Weaknesses:**

1. **(Clarity of the work)**. The paper is meandering and very hard to read. It introduces a lot of quantities that are poorly motivated without properly formalizing their definition and without even providing enough intuitions to justify their necessity. While the experimental section is more linear and easier to understand and interpret, the previous 2 sections (3 and 4) are the weakest part of the paper. I’d suggest the authors to revise the whole paper and make the hypothesis being tested clearer, as well as the main ideas that lead to the proposed method and the necessity of each design choice.
2. **(Identifiability of the latent factors of variation)**. Not having any guarantees on the identifiability of the latent variables jeopardizes the proposed method, which seeks to find meaningful/interpretable latent variables without any explicit supervision. When is a given amount of data enough to guarantee that the proposed method can recover all the latent factors? And, are there any other specific requirements needed to have disentangled factor of variations as discussed in [2]?
    - Since no theoretical guarantee is provided, it would be good to contextualize more the empirical claim on the motif identifiability which in turn, will justify the use of the proposed method.

References:

[1] M. Fumero, et al. “Leveraging sparse and shared feature activations for disentangled representation learning”

[2] F. Locatello, et al. “Weakly-Supervised Disentanglement Without Compromises”

**Questions:**

- Can the authors comment more the sparsity assumption on g^* presented in Section 3.1?
- Section 3.3 is hard to parse and does not clearly highlight the connection between the information bound and SPARLING.



Minor:
- The paper contains some typos, I suggest the authors to proofread the manuscript.

---

> ### Author Response · Authors · 2023-11-22
>
> > (Clarity of the work). The paper is meandering and very hard to read. It introduces a lot of quantities that are poorly motivated without properly formalizing their definition and without even providing enough intuitions to justify their necessity. While the experimental section is more linear and easier to understand and interpret, the previous 2 sections (3 and 4) are the weakest part of the paper. I’d suggest the authors to revise the whole paper and make the hypothesis being tested clearer, as well as the main ideas that lead to the proposed method and the necessity of each design choice.
>
> Thank you for the helpful feedback, we'll do our best to improve the presentation.
>
> > (Identifiability of the latent factors of variation). Not having any guarantees on the identifiability of the latent variables jeopardizes the proposed method, which seeks to find meaningful/interpretable latent variables without any explicit supervision. When is a given amount of data enough to guarantee that the proposed method can recover all the latent factors? And, are there any other specific requirements needed to have disentangled factor of variations as discussed in [2]?
>
> This paper is empirical in nature and validates its empirical hypothesis in three different domains. While it would of course be better to have a theoretical result as well, we believe that it is out of the scope of the paper to provide one. A theoretical guarantee would also be inherently limited to the kinds of mathematical objects that could be easily analyzed within a theoretical framework, whereas here we demonstrate that the technique works empirically on a g network that takes the form of a multi-layer residual convolutional network and an h network that is a complex LSTM/transformer architecture.
>
> > Since no theoretical guarantee is provided, it would be good to contextualize more the empirical claim on the motif identifiability which in turn, will justify the use of the proposed method.
>
> It is not completely clear what is meant by “contextualizing” in this case. Should we understand this as making more explicit early on in the paper that this is an empirical claim and making clear early on the specific instances where it has been validated? Or is it about providing more guidance to the reader as to what kinds of practical situations the approach might be useful for? If it is the former, we are happy to do so. If the latter, this is new work and we are in the process of investigating potential applications to more realistic problems. As mentioned in https://openreview.net/forum?id=zgHamUBuuO&noteId=tJhJMRIRWh , a variant of Sparling has been used in a biological application.
>
> > Can the authors comment more the sparsity assumption on g^* presented in Section 3.1?
>
> The sparsity assumption refers to the assumption that the true g^* of the data-generating process satisfies the property of sparsity, that is E_{x ~ D}[NZ(g^*(x))] < delta for some small delta, where we define NZ(m) to be the fraction of elements in the tensor m that are nonzero. This assumption can be thought of as restricting the range of g^* to a subset of its codomain.
>
> > Section 3.3 is hard to parse and does not clearly highlight the connection between the information bound and SPARLING.
>
> We have rewritten the ending of this section to better highlight the connection. Let us know if you would like further clarification.
>
> > The paper contains some typos, I suggest the authors to proofread the manuscript.
>
> We have done so.

---

### Official Review · Reviewer_eQb3 · 2023-11-01

**Soundness:** 3 good
**Presentation:** 4 excellent
**Contribution:** 3 good
**Rating:** 5
**Confidence:** 4

**Summary:**

The paper proposes a technique to learn extremely-sparse intermediate representations without any additional supervision on the representation. The proposed method is to set activations below an annealed threshold value to 0 during training. The paper shows via experiments on 3 datasets that Sparling is able to learn sparse representations without meaningful decrease in task error, unlike considered benchmarks. Learned representations are reliably linkable to known motifs that are present in each of the considered datasets.

**Strengths:**

* The problem of effectively learning sparse representations is important and the proposed approach is effective and novel to my knowledge.
* The empirical results presented on selected datasets are impressive, especially compared to baselines shown.
* The proposed method is extremely simple, and does not require any additional supervision on the representation.

**Weaknesses:**

* The experiments in the paper are limited to settings where strong locality priors may be used. It is unclear if the method works in more general settings, and this limits how significant it is. Can Sparling be applied in standard image classification tasks to learn sparse but predictive features?
* The baselines considered are not totally fair. While the Sparling coefficient $t$ is annealed during the training process to alleviate optimization challenges, the coefficient used for L1 loss is set only once, likely preventing L1 loss from learning sparse representations due to the same optimization challenges. Could we see ablations where the L1 coefficient is also annealed similar to the Sparling coefficient? In particular, is the improved sparsity a result of the spatial sparsity layer, or just the annealing scheme?

**Questions:**

* Does Sparling work for non-convolutional architectures (e.g. MLP)?
* In Figure 6, the retrained $\hat{h}$ outperforms the end-to-end non sparse network. How is this the case? Why is the performance different at all from the Sparling network which trains the motif model and prediction head end-to-end?
* What is the motivation for computing the spatial sparsity layer channelwise?

---

> ### Author Response · Authors · 2023-11-22
>
> > The experiments in the paper are limited to settings where strong locality priors may be used. It is unclear if the method works in more general settings, and this limits how significant it is. Can Sparling be applied in standard image classification tasks to learn sparse but predictive features?
>
> See https://openreview.net/forum?id=zgHamUBuuO&noteId=tJhJMRIRWh for our response.
>
> > The baselines considered are not totally fair. While the Sparling coefficient t is annealed during the training process to alleviate optimization challenges, the coefficient used for L1 loss is set only once, likely preventing L1 loss from learning sparse representations due to the same optimization challenges. Could we see ablations where the L1 coefficient is also annealed similar to the Sparling coefficient? In particular, is the improved sparsity a result of the spatial sparsity layer, or just the annealing scheme?
>
> An initial analysis, using an exponentially annealing L1 parameter (specifically, starting at 0.1 and increasing by 33% every time accuracy hits the threshold), shows that this performs slightly less well than our MT technique (about 5% error vs 1.7-3.8% for ours). We will analyze this more completely with different annealing schedules in future, but do not have sufficient time to properly implement this for this particular rebuttal cycle. Even if this were to work, it would not allow us to target specific sparsity values as easily (e.g., if you for one reason or another wanted to consistently keep two parts of your input at different sparsities), and it would require an additional hyperparameter to tune in the form of the initial L1 loss coefficient.
>
> (To clarify, in Sparling, t is not annealed during the training process, rather it is targeted to a quantile of the output of the g network, specified by delta; delta is annealed.)
>
> > Does Sparling work for non-convolutional architectures (e.g. MLP)?
>
> Sparling requires some notion of locality for the g layer, but the h layer can be arbitrary. In our case, we used transformers because we were producing sequential outputs, but this is not a necessary feature of the system. In future work, we will explore different h architectures.
>
> > In Figure 6, the retrained h outperforms the end-to-end non sparse network. How is this the case?
>
> We do not believe that this apparent improvement reflects a genuine advantage to the Retrained technique over end-to-end training. The small difference that can be seen in the diagram is not statistically significant, and we have clarified this in the paper. The numerical difference is also small enough to be explained by minor technical details of convergence criterion in training. In the paper we do not claim Retrained is better than Non-Sparse, merely that it is not worse in two domains, and only slightly worse in the third.
>
> > Why is the performance different at all from the Sparling network which trains the motif model and prediction head end-to-end?
>
> In the Sparling setting, we are imposing an information bottleneck to force g to have semantically meaningful outputs. The same g is used in the Retrained setting, but since the bottleneck is removed, the network h is able to take advantage of the additional information that would not have been allowed past the bottleneck.
>
> This becomes relevant because of the presence of noise. As there is noise, there are often motifs that might be real or might be noise, just at the threshold of detection. In a model with no bottleneck, the g model can pass information about these potential motifs through and disambiguate later in the process. However, in the presence of a strong sparsity bottleneck, the model cannot do this without passing through too many nonzero activations. As we describe in the paper, this requires the model “commit” to choosing one or the other. This results in degraded performance.
>
> > What is the motivation for computing the spatial sparsity layer channelwise?
>
> There are two alternatives that come to mind.
>
> The first, using a threshold at each position and channel, would result in a very large number of thresholds and be quite noisy. We make the assumption of homogeneity across positions in the input, as do most techniques dealing with 1d or 2d inputs.
>
> The second, using a single threshold across all channels, is something we did explore. You can find our analysis in Appendix D. A summary is that this approach leads to substantially higher confusion error on average, while leading to lower false positive error. Confusion error is the error we are most concerned with, so we decided to use per-channel threshold setting.

---

### Official Review · Reviewer_M5Zp · 2023-11-01

**Soundness:** 2 fair
**Presentation:** 2 fair
**Contribution:** 2 fair
**Rating:** 3
**Confidence:** 3

**Summary:**

The paper presents an algorithm to learn sparse intermediate representations through constraining activations to be sparse. The proposed algorithm relies on a combination of a ``Spatial Sparsity layer'' and ``adaptive sparse training". The spatial sparsity layer leverages a parametric form of ReLU to control sparsity. The adaptive sparse training anneals the sparsity parameter to encourage learning. The algorithm is evaluated on motif prediction and localization on three datasets.

**Strengths:**

1. Sparling achieves extreme activation sparsity and promotes learning interpretable representations.
2. The algorithm shows good localization performance on the provided evaluation datasets.
3. The idea of using activation sparsity to maintain representation capacity while allowing for sparse interpretable representations is interesting and could be an interesting direction to study for other models.

**Weaknesses:**

1. The requirement of $g^*$ being necessary for the final prediction is quite harsh for general settings. While this might be suitable for OCR, it is fairly rare to have localized and independent predictive features in an input.

2. The evaluations are limited to OCR style tasks, and audio detection. However, the paper is missing comparisons with other general OCR methods, including Deng (2016) which has been referred to as inspiring several design choices. In addition, the paper suggests several possible applications and downstream tasks, but does not tackle them. I suggest that the authors add more evaluations and comparisons with OCR methods, and well as end-to-end evaluations on im2Latex to begin, and possibly address other downstream tasks like neural attribution as well.

3. The writing (especially figure captions) can be made more clear.

**Questions:**

Could you elaborate Fig 2? Do the letters represent specific shapes or the digits themselves?

---

> ### Author Response · Authors · 2023-11-22
>
> > The requirement of g* being necessary for the final prediction is quite harsh for general settings. While this might be suitable for OCR, it is fairly rare to have localized and independent predictive features in an input.
>
> See https://openreview.net/forum?id=zgHamUBuuO&noteId=tJhJMRIRWh for our response.
>
> > The evaluations are limited to OCR style tasks, and audio detection. However, the paper is missing comparisons with other general OCR methods, including Deng (2016) which has been referred to as inspiring several design choices. In addition, the paper suggests several possible applications and downstream tasks, but does not tackle them. I suggest that the authors add more evaluations and comparisons with OCR methods, and well as end-to-end evaluations on im2Latex to begin, and possibly address other downstream tasks like neural attribution as well.
>
> Our goal in this work was not to improve end-to-end accuracy, but rather to demonstrate how an architectural change to a neural network could be used to leverage end-to-end data to implicitly learn an intermediate step in the mechanism without the need for any supervision. As such, we did not compare to alternate end-to-end models, as this is orthogonal to the goal of the paper. When we mention Deng is used as inspiration for design choices in this paper, we are primarily referring to the architecture of the g and h network, rather than the core aspects of the technique we are advancing.
>
> > The writing (especially figure captions) can be made more clear.
>
> We have taken a pass over the paper and clarified/elaborated the captions. If you have any more specific feedback, we would be happy to incorporate it.
>
> > Could you elaborate Fig 2? Do the letters represent specific shapes or the digits themselves?
>
> Each letter represents a separate channel of the post-Sparse layer of our neural network. For example, in the DigitCircle example, the A channel represents the first channel out of 10, the C channel represents the third, the D channel the fourth, etc. In practice, for this network, once trained, you can think of the C channel as representing the 6 digit: however, this is an emergent property of the network rather than something that it was specifically trained to do. This property, where the channels end up recognizing the true intermediate features, is the main claim we are trying to establish with our experiments. I have updated the caption to clarify this.

---

> > ### Comment · Reviewer_M5Zp · 2023-11-22
> > **Thanks for the Response**
> >
> > Thank you for the clarifications and the response. The modified captions improve the presentation of the paper quite a bit. I advise the authors to increase the contrast on Fig 2 if possible to improve visibility.
> >
> > With respect to comparisons, while the end to end performance may not be a contribution of the paper, it is necessary to evaluate if Sparling loses anything by introducing sparsity in the architecture. Not only will the comparisons help clarify any caveats to using Sparling vs other methods, it will also showcase how it adds interpretability over approaches like Deng et al. I disagree with the authors' claim here that evaluations are orthogonal to the message in the paper.

---

### Author Response · Authors · 2023-11-22
**Requirements on the Domain and g^***

We believe that all our requirements on g^* are reasonable when considering the kinds of domains we state Sparling is a good fit for. The sparsity and locality assumptions arise naturally in domains where there is a large object under analysis, but only some aspects of it are relevant to a downstream prediction task. This comes up in situations such as OCR or listening to audio for specific words, as well as scientific applications. For example, some work has been performed using a modification of Sparling on the prediction of RNA splice sites [1]. RNA has the properties described above, in that it contains dense information of many kinds, but the information related to splicing is sparsely distributed throughout the sequence in small localized regions (patterns of RNA bases that proteins can attach to).

The necessity requirement, while it seems harsh at first, is only really a requirement of the modeled causal process rather than a requirement of the domain. As seen with the LaTeX example and parentheses, if the necessity requirement is not met, this does not lead to a loss of performance of the technique on necessary features, it merely leads to the unnecessary features not being recognized properly by the system. This affects the error metric, but not the performance of the technique on the necessary features. In the case of RNA, for example, one might think of the necessary features as the sites of proteins that are involved in splicing, and unnecessary features being things like sequence elements that are responsible for protein coding, or the sites of protein binding for proteins not involved in RNA splicing.

[1] (WARNING: this BioRXiV paper contains a citation to an unblinded ArXiV version of the paper you are reviewing) Improved modeling of RNA-binding protein motifs in an interpretable neural model of RNA splicing. Kavi Gupta, Chenxi Yang, Kayla McCue, Osbert Bastani, Phillip A Sharp, Christopher B Burge, Armando Solar-Lezama. bioRxiv 2023.08.20.553608; doi: https://doi.org/10.1101/2023.08.20.553608

---

### Meta-Review · Area_Chair_iHKF · 2023-12-05

**Metareview:**

This work introduces a method to achieve sparse hidden activations, by leveraging an annealed spatially-aware thresholding scheme. The authors demonstrate the efficacy and interpretability of their method on multiple synthetic datasets.

This paper presents a thought-provoking approach and viewpoint on "good properties" of intermediate representations in neural nets.

However, the paper has several shortcomings, most importantly a lack of real-world problems, insufficient comparison to strong baselines and lacks motivation. Reviewers agree that the work is not ready to be published at ICLR in its current form.

**Justification For Why Not Higher Score:**

The work had too many shortcomings to meet the bar of an ICLR publication, all reviewers agreed on this.

**Justification For Why Not Lower Score:**

There is no lower score.

---

### Decision · Program_Chairs · 2024-01-16

Reject